# Supervised Graph Contrastive Learning for Gene Regulatory Network

## Abstract

Graph representation learning is effective for obtaining a meaningful latent space utilizing the structure of graph data and is widely applied, including biological networks. In particular, Graph Contrastive Learning (GCL) has emerged as a powerful self-supervised method that relies on applying perturbations to graphs for data augmentation. However, when applying existing GCL methods to biological networks such as Gene Regulatory Networks (GRNs), they overlooked meaningful biologically relevant perturbations, e.g., gene knockdowns. In this study, we introduce SupGCL (Supervised Graph Contrastive Learning), a novel GCL method for GRNs that directly incorporates biological perturbations derived from gene knockdown experiments as the supervision. SupGCL mathematically extends existing GCL methods that utilize non-biological perturbations to probabilistic models that introduce actual biological gene perturbation utilizing gene knockdown data. Using the GRN representation obtained by our proposed method, our aim is to improve the performance of biological downstream tasks such as patient hazard prediction and disease subtype classification (graph-level task), and gene function classification (node-level task). We applied SupGCL on real GRN datasets derived from patients with multiple types of cancer, and in all experiments SupGCL achieves better performance than state-of-the-art baselines.

## 1 Introduction

Graph representation learning has recently attracted attention in various fields to learn a meaningful latent space to represent the connectivity and attributes in given graphs [1]. Applications of graph representation learning are advancing in numerous areas where network data exists, such as analysis in social networks, knowledge graphs [2, 3], and biological network analysis in bioinformatics [4, 5].

Among these applications, the use of graph representation learning for Gene Regulatory Networks (GRNs), where each node and edge represents important intracellular functions and/or processes, is particularly significant in the fields of biology and drug discovery, as it is expected to contribute to identifying therapeutic targets and understanding disease mechanisms. GRN representation learning has been applied to tasks such as inferring transcription factors [6] and predicting drug responses in cancer cell lines [7].

With advancements in gene expression measurement and analysis technologies, identification methods for GRNs from expression data are also evolving. Traditional GRN identification constructs networks using statistical techniques applied to patient populations. Recently, it has become possible to construct GRNs specific to individual patients, highlighting distinct gene regulatory patterns compared to the population as a whole [8]. Hereafter in this paper, we refer to such individualized networks simply as GRNs. Similarly, in cell-based experiments such as gene knockdowns, it is now possible to estimate distinct GRNs for each experiment depending on the expression profile.

Submitted to 39th Conference on Neural Information Processing Systems (NeurIPS 2025). Do not distribute.

Graph representation learning applied to GRNs is believed to be effective for a wide range of biological applications. Among such methods, Graph Contrastive Learning (GCL) has gained traction. GCL enhances graph data via artificial perturbations applied to nodes or edges, and learns useful graph embeddings by maximizing the similarity between differently augmented views of the same graph [9]. For example, we can perform contrastive learning by artificial perturbations such as randomly removing nodes to augment the GRN and consequently learn a representation for the GRN.

However, a major challenge arises in that the artificial perturbations employed in conventional GCL methods significantly deviate from genuine biological perturbations, making it difficult to learn effective representations when applied to GRNs. A related issue occurs in heterogeneous node networks, where random perturbations to nodes or edges during augmentation can disrupt topology and attribute integrity, ultimately hindering representation learning [10, 11]. This problem is especially relevant to GRNs, where node heterogeneity exists—for example, in the presence of master regulators [12].

To address these issues, we propose a novel supervised GCL method (SupGCL) that leverages gene knockdown perturbations within GRNs. Our method uses experimental data from actual gene knockdowns as supervision, enabling biologically faithful representation learning. In gene knockdown experiments, the expression of specific genes is suppressed, representing biological perturbations that allow for the inference of GRNs. By using these perturbations as supervision signals for GCL, we can perform data augmentation that retains biological characteristics. Moreover, since our method naturally extends traditional GCL models in the direction of supervised augmentation within a probabilistic framework, conventional GCL approaches emerge as special cases of our proposed model.

To evaluate the effectiveness of the proposed SupGCL method, we apply it to GRN datasets from cancer patients across three cancer types and conduct multiple downstream tasks. For gene-level downstream tasks, we perform classification into Biological Process, Cellular Component, and cancer-related gene categories. For patient-level tasks, we conduct hazard prediction and disease subtype classification. The performance of our method is compared against existing graph representation learning techniques, including conventional GCL methods.

The main contributions of this study are as follows:

- **Proposal of a novel GRN representation learning method utilizing gene knockdown experiments:** We develop a new GCL method tailored for GRNs that incorporates gene knockdown data as supervision to enhance biological plausibility.

- **Theoretical extension of GCL:** We formulate supervised GCL, incorporating augmentation selection into a unified probabilistic modeling framework, and theoretically demonstrate that existing GCL methods are special cases of our proposed approach.

- **Empirical validation of the proposed method:** We apply the method to 13 downstream tasks on GRNs derived from real cancer patients and consistently outperform conventional approaches across all tasks.

Our implementation and all experimental codes are available on http://github.com/xxxxxx.

## 2 Related Work

Graph Contrastive Learning (GCL) has inspired the development of numerous methods, largely based on the design of data augmentations and the construction of positive/negative sample pairs [10]. GCL methods can be broadly categorized into three types according to how they generate these training pairs: (1) graph-level pairs, (2) node-level pairs, and (3) cross-model pairs.

A representative method using graph-level pairs is GraphCL [13], which applies random data augmentations to graphs and treats the resulting two graph views as a positive pair. This approach enhances the model's ability to capture graph-level representations. However, it has been pointed out that node-level information can become obscured in the process [14]. This limitation is especially problematic in applications like Gene Regulatory Networks (GRNs), where the semantics of individual nodes are crucial.

In contrast, methods such as GRACE, which generate node-level pairs, apply augmentations to graphs and treat embeddings of different nodes as negative pairs during training [15]. This enables more precise representation learning that captures local structure and attribute information at the node level. Although GRACE has been applied to real GRNs [6], the augmentation strategies used do not incorporate the specific biological characteristics of GRNs.

Beyond direct graph manipulations, methods like BGRL [16] generate positive pairs across models—between two instances of the same graph embedding model operating at different learning speeds—rather than relying on heuristic graph augmentations. Such cross-model pairing strategies have attracted attention for their ability to learn graph representations in a more natural manner. Notably, the recently proposed SGRL [17] achieves stable and high-performance self-supervised learning by using a pair of models: one that distributes node embeddings uniformly on a hypersphere, and another that incorporates graph topology information. However, these methods are constrained by their inability to design task-specific perturbations, and their applicability to biological networks such as GRNs remains unexplored and unverified.

# 3 Preliminaries

## 3.1 Background of Graph Contrastive Learning

Although there are various definitions of contrastive learning, it can be expressed using a probabilistic model based on KL divergence over pairs of augmentations or node instances [18]. Let $\mathcal{X}$ denote a set of entities and let $(x, y) \in \mathcal{X} \times \mathcal{X}$ be a pair from that set. The contrastive loss is formulated as follows:

$$\text{Loss}_{\text{I-CON}} \triangleq \frac{1}{|\mathcal{X}|} \sum_{x \in \mathcal{X}} D_{\text{KL}}(p_\theta(y|x) | q_\phi(y|x)). \tag{1}$$

Here, $q_\phi(y|x)$ is the probability distribution of the target model with parameter $\phi$, and $p_\theta(y|x)$ is a reference distribution. To avoid trivial solutions when training both $p_\theta$ and $q_\phi$ simultaneously, the reference distribution $p_\theta$ is almost fixed. The reference model $p_\theta(y|x)$ is often designed as a probability that assigns a non-zero constant to positive pairs $(x, y)$ and zero to negative pairs $(x, y)$.

Graph Contrastive Learning (GCL) handles the target model $q_\phi(j|i)$ corresponding to a pair of nodes $(i, j)$. Consider graph operations for augmentation, order them, and represent the index of these operations by $a$. Let $z_i^a \in \mathbb{R}^d$ be the graph embedding of the $i$-th node obtained from the Graph Neural Network under the $a$-th augmentation operation. For two augmentation operations $(a, b)$, the pair of probability models $(p, q_\phi^{a,b})$ used in GCL is defined by

$$p(j|i) \triangleq \delta_{ij}, \quad q_\phi^{a,b}(j|i) \triangleq \frac{\exp\big(\text{sim}(z_i^a, z_j^b)/\tau_{\text{n}}\big)}{\sum_{k \in \mathcal{V}} \exp\big(\text{sim}(z_i^a, z_k^b)/\tau_{\text{n}}\big)}. \tag{2}$$

Here, $\mathcal{V}$ is the set of nodes in the given graph, $\delta_{ij}$ is the Dirac delta, $\tau_{\text{n}} > 0$ is a temperature parameter and $\text{sim}(\cdot, \cdot)$ denotes cosine similarity. This setting is often extended so that the definitions of $(p, q_\phi^{a,b})$ vary according to how positive and negative pairs are sampled. Note that the target model $q_\phi$ depends on the sampling method of augmentation operators, so the probability model also depends on $(a, b)$.

GCL trains the model using the following loss function on the pair of probability models $(p, q_\phi^{a,b})$ induced by augmentation operations $(a, b)$, according to the formulation of contrastive learning loss (1).

$$\text{Loss}_{\text{node}}^{a,b} \triangleq \frac{1}{|\mathcal{V}|} \sum_{i \in \mathcal{V}} D_{\text{KL}}(p(j|i) | q_\phi^{a,b}(j|i)). \tag{3}$$

This encourages the embeddings at the node level $z_i^a$ and $z_i^b$ of the same node under different augmentation operations to be close to each other. Typically, augmentation operations $a, b$ are chosen by uniform sampling from a set of candidates $\mathcal{A}$. Hence, in practice, the expected value is minimized under the uniform distribution $\text{U}_\mathcal{A}$ over $\mathcal{A}$:

$$\text{Loss}_{\text{node}} \triangleq \mathbb{E}_{a,b \sim \text{U}_\mathcal{A}}[\text{Loss}_{\text{node}}^{a,b}]. \tag{4}$$

127 While GCL achieves node-level representation learning via the procedure described above, in many
128 cases the augmentation operations themselves rely on artificial perturbations such as randomly adding
129 and/or deleting nodes and/or edges. In this study, we introduce gene knockdown—a biological
130 perturbation—as supervision for these augmentation operations.

## 3.2 Notation and Problem Definition

132 In this study, we describe a GRN as a directed graph $\mathcal{G} \triangleq (\mathcal{V}, \mathcal{E}, X^{\mathcal{V}}, X^{\mathcal{E}})$ that contains information
133 on nodes and edges. Here, $\mathcal{V}$, and $\mathcal{E}$ are the sets of nodes and edges, respectively, and each node
134 represents a gene. $X_i^{\mathcal{V}}$ is the feature of the $i$-th gene, and $X_i^{\mathcal{E}} \in \mathbb{R}^{|\mathcal{E}|}$ is the feature of the $i$-th edge
135 in the network. The augmentation operation corresponding to the knockdown of the $i$-th gene is
136 modeled by setting the feature of the $i$-th gene to zero and also setting the features of all edges
137 connected to the $i$-th gene to zero.

138 We associate the $a$-th augmentation operation with the knockdown of the $a$-th gene. In what follows,
139 we denote by $\mathcal{G}_a$ the graph obtained by applying the $a$-th augmentation operation to $\mathcal{G}$. Moreover, in
140 this study, let $\mathcal{H}_a$ be the teacher GRN for the knockdown of the $a$-th gene , and let $\mathcal{K}$ be the set of
141 all augmentation operations for which such teacher GRNs exist. In other words, $\mathcal{H}_a$ is a GRN that
142 serves as a teacher for artificial augmentation for the $a$-th gene.

143 Our goal is to use the original GRN $\mathcal{G}$ and its teacher GRNs $\{\mathcal{H}_a\}_{a \in \mathcal{K}}$ to train a Graph Neural
144 Network (GNN) $f_\phi$. Defining embedded representations through the GNN $f_\phi$ as

$$Z^a \triangleq f_\phi(\mathcal{G}_a) \in \mathbb{R}^{|\mathcal{V}| \times d}, \quad Y^a \triangleq f_\phi(\mathcal{H}_a) \in \mathbb{R}^{|\mathcal{V}| \times d}, \tag{5}$$

145 where $z_i^a$ and $y_i^a$ denote the embedding vectors of the $i$-th node in $Z^a$ and $Y^a$, respectively, and $d$ is
146 the embedding dimension. Note that the same GNN $f_\phi$ is used to produce both $Z^a$ and $Y^a$.

147 In this work, we train the neural network $f_\phi$ using the set of pairs $\{(Z^a, Y^a)\}_{a \in \mathcal{K}}$, where $(Z^a, Y^a)$
148 corresponds to the graph embedding obtained by the GRN augmentation operation and the embedding
149 of the teacher GRN for the corresponding gene knockdown.

## 4 Method

151 For the set of embedded representations $\{(Z^a, Y^a)\}_{a \in \mathcal{K}}$, we consider the pair of augmentation
152 operations $(a, b)$ and the pair of nodes $(i, j)$ according to the contractive learning scheme. First, for
153 the pair of augmentation operations $(a, b)$, we clarify the supervised learning problem for augmen-
154 tation operations using KL divergence and then propose SupGCL using a distribution over pairs
155 of combinations of nodes and extension operations. A sketch of the proposed method is shown in
156 Figure 1.

157 The probability distribution of augmentation operations is naturally introduced by using similarities in
158 the entire graph embedding space $\mathbb{R}^{|\mathcal{V}| \times d}$ (rather than per node). By introducing the Frobenius inner
159 product as the similarity in the matrix space, we define the probability models for the augmentation
160 operations as:

$$p_\phi(b|a) \triangleq \frac{\exp\left(\mathrm{sim}_F(Y^a, Y^b)/\tau_{\mathrm{a}}\right)}{\sum_{c \in \mathcal{K}} \exp\left(\mathrm{sim}_F(Y^a, Y^c)/\tau_{\mathrm{a}}\right)}, \quad q_\phi(b|a) \triangleq \frac{\exp\left(\mathrm{sim}_F(Z^a, Z^b)/\tau_{\mathrm{a}}\right)}{\sum_{c \in \mathcal{K}} \exp\left(\mathrm{sim}_F(Z^a, Z^c)/\tau_{\mathrm{a}}\right)}, \tag{6}$$

161 where $\mathrm{sim}_F(\cdot, \cdot)$ denotes the Frobenius inner product, and $\tau_{\mathrm{a}} > 0$ is a temperature parameter. Unlike
162 node-level learning, $p_\phi(b|a)$ is not a fixed constant but rather a reference distribution based on the
163 supervised embeddings $\{Y^a\}_{a \in \mathcal{K}}$. Both probability models $p_\phi$ and $q_\phi$ are parameterized by the same
164 GNN $f_\phi$.

165 Using these probability distributions, substituting the reference model $p_\phi(b|a)$ and the target model
166 $q_\phi(b|a)$ into the formulation of contrastive learning in (1) yields the loss function for augmentation
167 operations:

$$\mathrm{Loss}_{\mathrm{Aug}} \triangleq \frac{1}{|\mathcal{K}|} \sum_{a \in \mathcal{K}} D_{\mathrm{KL}}(p_\phi(b|a) | q_\phi(b|a)). \tag{7}$$

168 Minimizing this loss reduces the discrepancy in embedding distributions between the artificially
169 augmented graphs and the biologically grounded knockdown graphs. However, if both $p_\phi$ and $q_\phi$

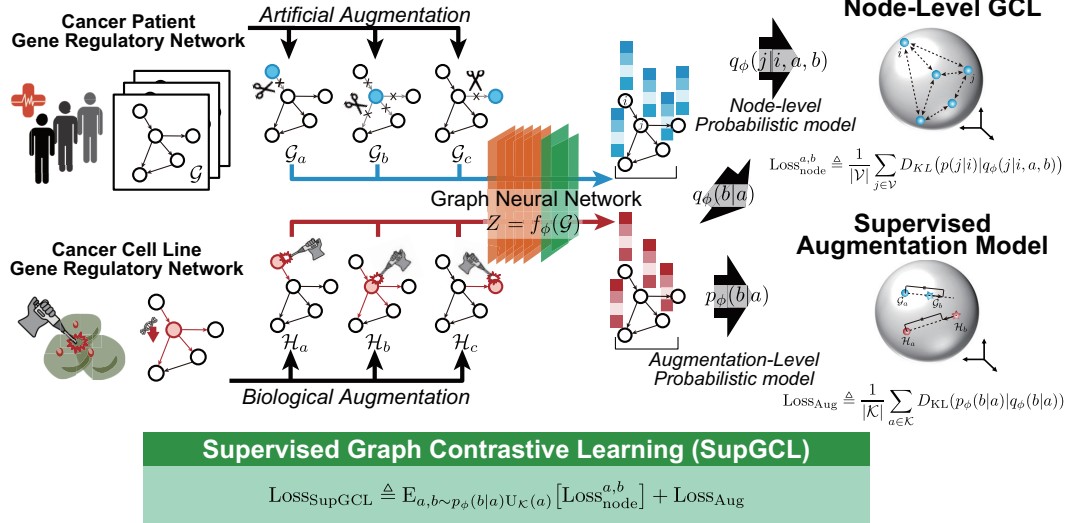

Figure 1: **Schematic overview of SupGCL (Supervised Graph Contrastive Learning).** The proposed method leverages two complementary types of graph augmentations. First, it generates artificially perturbed GRNs by simulating gene knockdowns, where node and edge features are masked based on the targeted gene. Second, it incorporates biologically grounded augmentations derived from real gene knockdown experiments conducted on cancer cell lines, serving as teacher GRNs. Embeddings are extracted using a shared GNN, and both node-level and augmentation-level contrastive losses are computed via KL divergence. This biologically grounded contrastive framework enables more faithful and effective representation learning of GRNs.

are optimized simultaneously, the model may converge to a trivial solution. For instance, if the GNN outputs constant embeddings, both distributions become uniform and $\mathrm{Loss}_{\mathrm{Aug}} = 0$. Thus, minimizing $\mathrm{Loss}_{\mathrm{Aug}}$ alone is insufficient for learning meaningful graph representations.

To address this issue, here we first introduce a reference model $p_\phi(j, b|i, a)$ and a target model $q_\phi(j, b|i, a)$ that use conditional probabilities for each pair of node and augmentation $(i, a), (j, b) \in \mathcal{V} \times \mathcal{K}$. By substituting these into the contrastive learning formulation in (1), we derive the loss function of Supervised Graph Contrastive Learning:

$$\mathrm{Loss}_{\mathrm{SupGCL}} \triangleq \frac{1}{|\mathcal{V}||\mathcal{K}|} \sum_{i \in \mathcal{V}, j \in \mathcal{K}} D_{\mathrm{KL}}(p_\phi(j, b|i, a)|q_\phi(j, b|i, a)). \tag{8}$$

Furthermore, the following theorem shows that by assuming independence between nodes and augmentation operations in the reference distribution $p_\phi$, we can avoid the trivial solution:

**Theorem 1.** *Assuming $p_\phi(i, j, a, b) = p(i, j)p_\phi(a, b)$, then*

$$\mathrm{Loss}_{\mathrm{SupGCL}} = \mathrm{E}_{a, b \sim p_\phi(b|a)\mathrm{U}_{\mathcal{K}}(a)}\big[\mathrm{Loss}_{\mathrm{node}}^{a,b}\big] + \mathrm{Loss}_{\mathrm{Aug}}. \tag{9}$$

**Proof:** This follows directly from the standard decomposition of KL divergence: $D_{\mathrm{KL}}(p(x, y)|q(x, y)) = \mathbb{E}_{x \sim p(x)}[D_{\mathrm{KL}}(p(y|x)|q(y|x))] + D_{\mathrm{KL}}(p(x)|q(x))$. See Appendix A for details. □

The first term in Theorem 1 corresponds to the expectation of the node-level GCL loss $\mathrm{Loss}_{\mathrm{node}}^{a,b}$ (as defined in Equation 3) with respect to the supervised augmentation distribution $p_\phi(b|a)$. This allows node-level contrastive learning to reflect biological similarity between knockdown operations. Importantly, since the theorem is independent of the specific choice of the node-level model $(p, q_\phi^{a,b})$, any contrastive loss described by KL divergence can be used in practice. Meanwhile, the second term reduces the distributional difference between the artificially generated augmentation-based GRN and

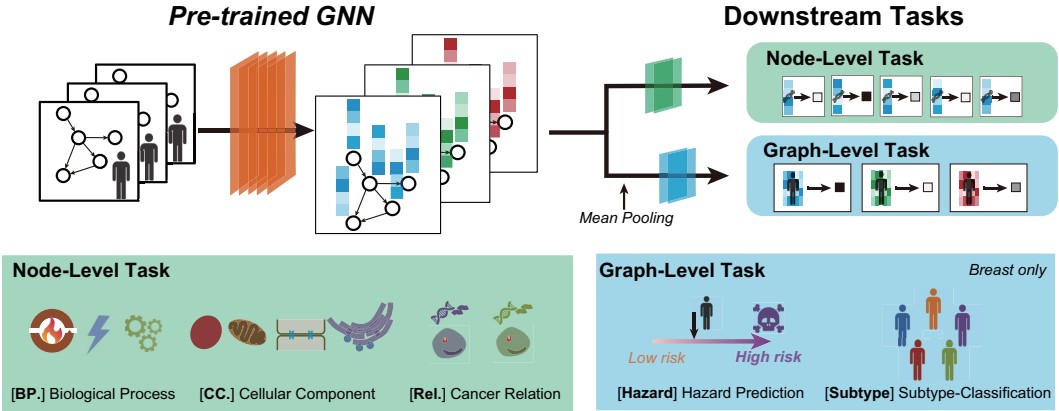

Figure 2: **Overview of Downstream Tasks Used for Benchmark of GRNs Across Three Cancer Types: Breast, Lung, and Colorectal Cancers.** The node-level tasks involve classifying genes into functional categories such as Biological Process [**BP.**], Cellular Component [**CC.**], and cancer relevance [**Rel.**]. The graph-level tasks include patient-level survival risk prediction [**Hazard**] and disease subtype classification [**Subtype**], the latter being specific to breast cancer. Mean pooling is applied to obtain graph-level representations.

Table 1: Description of downstream task

| Task | Task Type | Metrics |
|------|-----------|---------|
| **Node-Level Task** | | |
| [**BP.**]: Biological process classification | Multi-label binary classification (with 3 labels) | Subset accuracy |
| [**CC.**]: Cellular component classification | Multi-label binary classification (with 4 labels) | Subset accuracy |
| [**Rel.**]: Cancer relation | Classification (binary) | Accuracy |
| **Graph-Level Task** | | |
| [**Hazard**]: Hazard prediction | Survival analysis (1-dim risk score) | C-index |
| [**Subtype**]: Disease subtype prediction | Classification (5 groups) | Subtype accuracy |

the teacher GRN. Together, these two components ensure both expressive node representations and biologically meaningful augmentations.

Moreover, the performance of node-level representation learning and the biological validity following the teacher data for augmentation operations can be controlled by the temperature parameters $\tau_{\mathrm{n}}, \tau_{\mathrm{a}}$ of each probability model. In particular, when the temperature parameter $\tau_{\mathrm{a}}$ involved in the augmentation operation is sufficiently large, the augmentation operation becomes independent of the teacher GRNs $\{Y_a\}_{a \in \mathcal{K}}$, and coincides with the conventional node-level GCL loss function.

**Corollary 1.** $\lim_{\tau_{\mathrm{a}} \to \infty} \mathrm{Loss}_{\mathrm{SupGCL}} = \mathrm{Loss}_{\mathrm{node}}$.

**Proof:** As $\tau_{\mathrm{a}} \to \infty$, we have $p_\phi(b|a) \to \mathrm{U}_{\mathcal{K}}(b)$ and $q_\phi(b|a) \to \mathrm{U}_{\mathcal{K}}(b)$. Therefore, the expectation term becomes: $\lim_{\tau_{\mathrm{a}} \to \infty} \mathrm{E}_{a,b \sim p_\phi(b|a)\mathrm{U}_{\mathcal{K}}(a)}\big[\mathrm{Loss}_{\mathrm{node}}^{a,b}\big] = \mathrm{E}_{a,b \sim \mathrm{U}_{\mathcal{K}}}[\mathrm{Loss}_{\mathrm{node}}^{a,b}]$, $\lim_{\tau_{\mathrm{a}} \to \infty} D_{\mathrm{KL}}(p_\phi(b|a)|q_\phi(b|a)) = 0$ thus proving the corollary. $\square$

In this study, we train the GNN using standard gradient-based optimization applied to the loss function defined in Theorem 1. The corresponding pseudocode is provided in Appendix B.

## 5 Experiments

In this study, we formulated SupGCL by naturally extending the loss function of conventional GCL, based on a contrastive learning framework using KL divergence. Furthermore, we clarified the relationship between SupGCL and conventional GCL through the temperature parameter. This chapter verifies the effectiveness of the proposed method using actual gene regulatory networks (GRNs) from cancer patients and augmented GRNs based on gene knockdown experiments.

Table 2: Finetuning result of node-level downstream task.

| Task | w/o-pretrain | GAE | GraphCL | GRACE | SGRL | SupGCL |
|---|---|---|---|---|---|---|
| **BP.** | | | | | | |
| Breast | 0.232±0.031 | 0.230±0.029 | 0.167±0.042 | 0.230±0.051 | 0.220±0.052 | **0.243±0.052** |
| Lung | 0.259±0.056 | 0.247±0.038 | 0.115±0.024 | 0.259±0.063 | 0.233±0.027 | **0.282±0.037** |
| Colorectal | 0.231±0.062 | 0.245±0.023 | 0.207±0.058 | 0.249±0.050 | 0.146±0.029 | **0.262±0.030** |
| **CC.** | | | | | | |
| Breast | 0.264±0.042 | 0.250±0.034 | 0.131±0.050 | 0.236±0.026 | 0.249±0.030 | **0.291±0.026** |
| Lung | 0.267±0.041 | 0.245±0.033 | 0.069±0.041 | 0.255±0.043 | 0.248±0.037 | **0.274±0.044** |
| Colorectal | 0.278±0.098 | 0.256±0.042 | 0.190±0.062 | 0.265±0.030 | 0.133±0.081 | **0.279±0.052** |
| **Rel.** | | | | | | |
| Breast | 0.573±0.033 | 0.561±0.059 | 0.553±0.051 | 0.575±0.035 | 0.580±0.055 | **0.600±0.057** |
| Lung | 0.575±0.053 | 0.568±0.029 | 0.555±0.036 | 0.592±0.038 | 0.593±0.034 | **0.604±0.053** |
| Colorectal | 0.563±0.071 | 0.574±0.049 | 0.535±0.056 | 0.576±0.071 | 0.580±0.042 | **0.594±0.039** |

Table 3: Finetuning result of graph-level downstream task.

| Task | w/o-pretrain | GAE | GraphCL | GRACE | SGRL | SupGCL |
|---|---|---|---|---|---|---|
| **Hazard** | | | | | | |
| Breast | 0.601±0.035 | 0.625±0.035 | 0.638±0.049 | 0.642±0.064 | 0.640±0.077 | **0.650±0.059** |
| Lung | 0.611±0.052 | 0.619±0.062 | 0.616±0.049 | 0.609±0.055 | 0.611±0.060 | **0.627±0.051** |
| Colorectal | 0.621±0.070 | 0.631±0.091 | 0.657±0.071 | 0.647±0.059 | 0.616±0.123 | **0.698±0.085** |
| **Subtype** | | | | | | |
| Breast | 0.804±0.031 | 0.834±0.028 | 0.719±0.077 | 0.841±0.026 | 0.829±0.030 | **0.847±0.036** |

## 5.1 Benchmark of Gene Regulatory Networks

**Evaluation Protocol:** We evaluated the proposed method through the following procedure. First, based on gene expression data from cancer patients, we constructed patient-specific GRNs. Similarly, we constructed teacher GRNs using gene knockdown experiment data. Then, pre-training was performed on the proposed method using both the patient-specific and teacher GRNs. Subsequently, the performance of the downstream tasks, such as classification accuracy and regression performance, was evaluated using the pre-trained models and compared against comparative methods. Finally, we visualized the latent representations at both the node and graph levels extracted from the trained models.

We compared the proposed method with the following five comparative models:

**w/o-pretrain** : Directly performs classification or regression for downstream tasks without any pre-training.

**GAE** : [19]: Graph representation learning method based solely on graph reconstruction.

**GraphCL** : [13]: Graph contrastive learning using positive pairs between graphs.

**GRACE** : [15]: Node-level graph contrastive learning method.

**SGRL** : [17]: Node-level GCL that leverages representation scattering in the embedding space.

**Datasets:** To evaluate the performance of SupGCL, we conducted benchmark evaluations using real-world datasets. For constructing patient-specific GRNs, we used cancer cell sample data from The Cancer Genome Atlas (TCGA). For constructing teacher GRNs, we used gene knockdown experiment data from cancer cell lines in the Library of Integrated Network-based Cellular Signatures (LINCS). The TCGA dataset [20] and the LINCS dataset [21] are both large-scale and widely-used public platforms providing gene expression data from cancer patients and cell lines, respectively.

We used normalized count data from the TCGA TARGET GTEx study [22] provided by UCSC Xena. For LINCS, we used normalized gene expression data from the LINCS L1000 GEO dataset (GSE92742) [23].

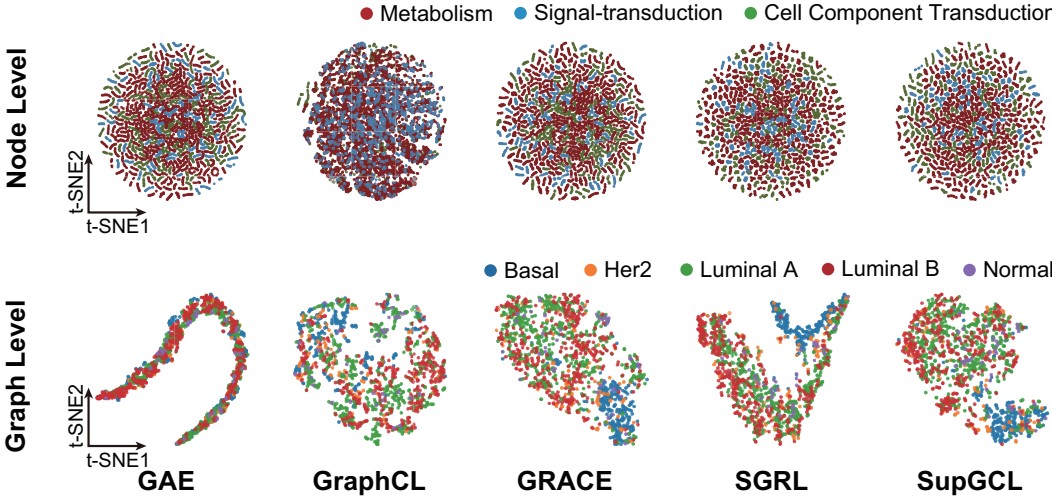

Figure 3: t-SNE visualization of pre-trained embeddings on breast cancer GRNs. The top row shows the node-level embedding space for individual genes, and the bottom row shows the corresponding graph-level readout features for each patient's network.

Experiments were conducted for three cancer types across both datasets: breast cancer, lung cancer, and colorectal cancer. Furthermore, the set of genes constituting each network was restricted to the 975 genes common to the TCGA gene set and the 978 LINCS landmark genes. The number of patient samples for each cancer type was N=1092 (breast), 1011 (lung), and 288 (colorectal), and the total number of knockdown experiments was 8793, 15926, and 11843, respectively. The number of unique knockdown target genes / total common genes was 768/975, 948/975, and 948/975.

The TCGA dataset also includes survival status and disease subtype labels associated with each gene expression profile. Additionally, each gene was annotated with multi-labels based on Gene Ontology [24] — Biological Process (metabolism, signaling, cell organization; 3 classes), and Cellular Component (nucleus, mitochondria, ER, membrane; 4 classes). We also used the OncoKB [25] cancer-related gene list to assign binary relevance labels. These labels were used for downstream tasks. Details are provided in Appendix C.

**Pre-processing:** To estimate the network structure of each GRN from gene expression data, we used a Bayesian network structure learning algorithm based on nonparametric regression with Gaussian noise [26]. For each experiment, gene expression values were used as node features, while edge features were defined as the product of estimated regression coefficients and the parent node's gene expression [27]. This structure estimation was performed per cancer type per dataset using the above algorithm. Further details are provided in Appendix D.

## 5.2 Result 1: Evaluation by Downstream Task

In this experiment, pre-training of the proposed and conventional methods was conducted using patient-specific GRNs from TCGA and teacher GRNs from LINCS. Subsequently, fine-tuning was performed on the pre-trained models using patient GRNs, and downstream task performance was evaluated (see Figure 2). During fine-tuning, two additional fully connected layers were appended to the node-level representations and graph-level representations (obtained via mean pooling), and downstream tasks were performed.

Graph-level tasks (hazard prediction, subtype classification) used survival and subtype labels from TCGA. Note that subtype classification was conducted only for breast cancer. Node-level tasks (Biological Process - BP, Cellular Component - CC, and cancer relevance - Rel.) used gene-level annotations from Gene Ontology and OncoKB.

Details of these downstream tasks are summarized in Table 1.

Each downstream task — hazard prediction, subtype classification, BP, and CC classification — was evaluated using 10-fold cross-validation. For cancer gene classification, due to label imbalance, we performed undersampling over 10 random seeds. Results are reported as mean ± standard deviation.

For all methods including the SupGCL and conventional methods, we used the same 5-layer Graph Transformer architecture [28]. Hyperparameters for pre-training were tuned with Optuna [29], and the model was optimized using the AdamW optimizer [30]. All training runs were performed on a single NVIDIA H100 SXM5 GPU. Additional experimental details can be found in Appendix E.

Tables 2 and 3 show the results for node-level and graph-level tasks, respectively. The best performance is indicated in bold, and the second-best is underlined. Although SupGCL did not achieve statistically significant superiority in every single task, it consistently outperformed other pre-training methods across all datasets and tasks.

For node-level tasks, many existing methods did not show much improvement over without-pretrain, whereas SupGCL consistently demonstrated strong performance. This suggests that SupGCL effectively captures biologically meaningful GRN representations suitable for these tasks. In graph-level tasks like hazard and subtype prediction, while some existing methods showed marginal improvement over without-pretrain, SupGCL achieved significantly higher performance.

### 5.3 Result 2: Latent Space Analysis

Using the pre-trained models, we visualized the embedding spaces derived from the breast cancer dataset (see Figure 3). Other similar visualizations are shown in Appendix F.

The top row of Figure 3 shows the node-level embeddings. Colors represent one of the three single-label Biological Process annotations (metabolism, signaling, or cell organization). Across all models, no clear clustering was observed based on labels. However, GraphCL's embeddings were notably different from others, with signs of latent space collapse. Detailed observations are provided in Appendix F. This suggests that GraphCL may be more suited to graph-level discrimination than node-level representation.

Compared to GraphCL, the other models showed more dispersed embeddings, though without distinct clustering.

The bottom row of Figure 3 illustrates the graph-level embeddings, colored by disease subtype. From this, it is evident that GAE and GraphCL fail to separate subtypes in the embedding space. In contrast, GRACE and SGRL showed moderate separation between the Basal subtype and others. SupGCL displayed the clearest separation, indicating its ability to better learn subtype-specific network representations.

## 6 Conclusion

In this study, we proposed a supervised graph contrastive learning method, SupGCL, for representation learning of gene regulatory networks (GRNs), which incorporates real-world genetic perturbation data as supervision during training. We formulated GCL with supervision-guided augmentation selection within a unified probabilistic framework, and theoretically demonstrated that conventional GCL methods are special cases of our proposed formulation. Through benchmark evaluations using downstream tasks based on both node-level and graph-level embeddings of GRNs from cancer patients, SupGCL consistently outperformed existing GCL methods.

A limitation of this paper is that the effectiveness of the proposed method has only been confirmed in situations where the teacher GRNs were constructed from knockdown experiments of the same cancer type.

As future work, we plan to expand the target cancer types and develop a large-scale, general-purpose SupGCL model that can operate across multiple cancer types.

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
