# OpenReview forum: "Supervised Graph Contrastive Learning for Gene Regulatory Network"
_NeurIPS.cc/2025/Conference — Submitted to NeurIPS 2025_

### Official Review · Reviewer_xKF9 · 2025-06-12

**Clarity:** 2
**Significance:** 2
**Originality:** 2
**Rating:** 3
**Confidence:** 3

**Summary:**

This paper focuses on gene regulatory networks and aims to pretrain a model that can be effectively adapted to various downstream tasks. Specifically, it transfers supervised graph contrastive learning methods to this domain. Experimental results demonstrate the usefulness of the pretraining approach.

**Questions:**

See above.

**Ethical Concerns:**

["NO or VERY MINOR ethics concerns only"]

**Final Justification:**

Thank you for your response. However, several of my concerns remain unresolved.

First, regarding the challenges mentioned in the manuscript, numerous augmentation-free methods for graph contrastive learning already exist. It is questionable whether the challenges you described are still present. What would be the result if these existing augmentation-free methods were directly applied to your task?

Second, the current state of research in gene regulatory networks is unclear in the manuscript, which makes it difficult to judge the novelty of your work.

Nevertheless, I appreciate your reply and have adjusted my score accordingly.

[1] Augmentation-Free Self-Supervised Learning on Graphs, AAAI2022

[2] SimGRACE: A Simple Framework for Graph Contrastive Learning without Data Augmentation, WWW2022

**Limitations:**

yes

**Quality:**

2

**Strengths And Weaknesses:**

Pros：

1.The study of gene regulatory networks is of significant scientific importance.


Cons：

1.The paper lacks novelty. It appears to explore a generic self-supervised graph representation learning method, as all the components used in the proposed approach are derived from existing and well-established graph contrastive learning (GCL) frameworks. Moreover, all baseline models used for comparison are classical GCL methods, with no analysis or comparison involving GRN-specific approaches. This makes it difficult to assess the fairness and significance of the reported performance gains, as the compared methods operate under different assumptions and levels of domain-specific modeling.

2.The related work section is insufficient. It lacks an in-depth discussion of prior research on GRNs, leaving readers unclear about the specific challenges in this domain. Additionally, the paper mentions that graph augmentations may alter the topology, but this issue has already been widely discussed and addressed in many prior works.

3.The manuscript contains redundant content in the background section. It is recommended to streamline this part and consider following the more concise formulation used in works like NT-Xent.

---

> ### Author Rebuttal · Authors · 2025-07-31
>
> Dear Reviewer,
>
> Thank you very much for your valuable and constructive feedback on our manuscript. We have carefully considered your comments and have revised our paper accordingly. We believe these revisions have significantly improved the quality and clarity of our work.
>
> Below, we address each of your concerns point-by-point.
>
> ---
>
> ### **Regarding Weakness 1: Novelty and Comparison with GCL**
>
> #### **A. Novelty and Distinction from General-Purpose GCL**
>
> First and foremost, we would like to emphasize a critical aspect of our work that distinguishes it from general-purpose Graph Contrastive Learning (GCL) frameworks. In our proposed method, the **supervisory signals**, namely the teacher graph $\mathcal{H}_a$ and its embedding $Y^a$, are **not generated from artificial data augmentations**. Instead, they are derived from **real gene expression data obtained from biological knock-down experiments**.
>
> The core novelty of our research lies in formulating a biologically and information-theoretically natural loss function that leverages these experimentally-derived teacher graphs. This is directly incorporated into our proposed loss function, $L_{SupGCL}$ (Eq. 9), specifically within the augmentation-related loss term $L_{Aug}$ and the expectation over the probability distribution $E_{p_\phi(b|a)}[\cdot]$. The teacher probability distribution $p_\phi(b|a)$ is calculated using the embedding $Y^a$ of the teacher graph $\mathcal{H}_a$ from actual knock-down experiments (Eq. 6).
>
> While our method is distinct from prior work in its use of real experimental data for supervision, we acknowledge that this specificity means it is not a universally applicable framework. Unlike general-purpose GCL methods like GraphCL or GRACE, our approach is applicable only to GRNs for which knock-down experimental data is available for pre-training.
>
> However, a key advantage of our approach is its practicality in downstream tasks. Both the teacher graph $H_a$ and the patient-specific graph $\mathcal{G}$ are embedded using the same GNN encoder $f_\phi$. This means that while the teacher graph from knock-down experiments is necessary during the pre-training phase, it is **not required for fine-tuning**. Consequently, fine-tuning can be performed using only the patient-specific graph $\mathcal{G}$ and the corresponding phenotype data. This is a significant departure from other omics-based prediction and representation learning models.
>
> #### **B. Comparison with GRN-Specific Methods**
>
> As you rightly pointed out, comparing our method with GRN-specific representation learning techniques is crucial. The field of GRN representation learning is advancing towards multimodal approaches, and as such, there are few highly specialized GNN architectures; comparisons are often made against general GNNs or graph feature methods (\[1\] Table 2).
>
> Among GRN-specific methods, **Gene2Vec \[2\]**, an application of Word2Vec to gene co-expression, is a widely used baseline. We have conducted **additional experiments** to compare our method against Gene2Vec, as well as against simpler baselines like a vanilla GAE and a standard GNN predictor, which are already included in our paper.
>
> The results below show that our method outperforms Gene2Vec in downstream tasks, confirming its effectiveness even when compared to GRN-specific approaches.
>
> | Method | Hazard Prediction (c-index, Breast Cancer) | Gene Function Prediction (Subset Accuracy, Breast Cancer) |
> | :--- | :--- | :--- |
> | Gene2Vec \[2\] | 0.602 ± 0.042 | 0.236 ± 0.041 |
> | **SupGCL (Ours)** | **0.650 ± 0.059** | **0.243 ± 0.052** |
>
> ---
>
> ### **Regarding Weakness 2: Related Work**
>
> #### **A. Prior Research on GRN Analysis**
>
> Thank you for this suggestion. We will expand the Related Work section to provide a more comprehensive overview of the GRN analysis landscape. GRN analysis methods can be broadly categorized into two main streams:
>
> 1.  **Network-inference methods**, which aim to infer edges (regulatory relationships) from expression data (e.g., GENIE3 \[3\]).
> 2.  **Representation-learning methods**, which learn transferable representations of the entire network for downstream tasks. This area is increasingly moving towards multimodal data, as seen in Muse-GNN \[1\].
>
> We will add this discussion to the Related Work section to better contextualize our study and clarify its position within the field.
>
> #### **B. Graph Representation Learning Considering Topological Impact**
>
> We agree that the topological impact of graph augmentations (e.g., node/edge dropping) is a well-studied problem in GCL. Research in this area includes:
>
> * Methods that avoid augmentation altogether, such as GAEs.
> * Methods that adaptively adjust augmentations to mitigate topological changes, such as GCA \[4\].
> * Methods that use two encoders with different parameters for contrastive learning, such as BGRL \[5\].
>
> In our manuscript, we already compare our method against GAEs. We also include SGRL, a successor to BGRL that achieved state-of-the-art results at NeurIPS 2024, as a key baseline.
>
> For completeness, we have also conducted an **additional experiment with GCA \[4\]**. The results, shown below, further confirm the superiority of our method. (We omitted a direct comparison with BGRL as it is effectively evaluated through its successor, SGRL).
>
> | Method | Hazard Prediction (c-index, Breast Cancer) | Gene Function Prediction (Subset Accuracy, Breast Cancer) |
> | :--- | :--- | :--- |
> | GCA \[4\] | 0.620 ± 0.039 | 0.241 ± 0.021 |
> | **SupGCL (Ours)** | **0.650 ± 0.059** | **0.243 ± 0.052** |
>
> ---
>
> ### **Regarding Weakness 3: Redundancy in Background**
>
> Thank you for pointing this out. Our formulation of the node-level loss, $\mathrm{Loss}_{\rm node}$, appears indirect because we adopted the unified contrastive learning framework defined by KL divergence, as proposed in I-CON.
>
> In practice, the loss function has a simpler, more direct form equivalent to the InfoNCE loss:
>
> $$
> \mathrm{Loss}_{\rm node} =  \mathrm{E}_{a,b\sim \mathrm{U}}\Big[\frac{1}{|\mathcal{V}|} \sum_{i\in \mathcal{V}}  \Big(-  \frac{1}{\tau_{\rm n}}\mathrm{sim}(z_i^a,z_i^b) +  \log\Big( \sum_{k\in  \mathcal{V}} \exp\Big(\frac{1}{\tau_{\rm n}}\mathrm{sim}(z_i^a,z_k^b)\Big)\Big) \Big)\Big]
> $$
>
> We will revise the manuscript to present the loss function more clearly, ensuring the core idea is easier to grasp, as you suggested.
>
> ---
>
> We hope that these revisions, clarifications, and additional experiments have adequately addressed your concerns. We thank you again for your time and effort in reviewing our manuscript.
>
> Sincerely,
> The Authors
>
> ### **References**
>
> \[1\] Liu, Tianyu, et al. "Muse-gnn: Learning unified gene representation from multimodal biological graph data." *Advances in Neural Information Processing Systems* 36 (2023).
>
> \[2\] Du, Jingcheng, et al. "Gene2vec: distributed representation of genes based on co-expression." *BMC genomics* 20.Suppl 1 (2019).
>
> \[3\] Huynh-Thu, Vân Anh, et al. "Inferring regulatory networks from expression data using tree-based methods." *PloS one* 5.9 (2010).
>
> \[4\] Zhu, Yanqiao, et al. "Graph contrastive learning with adaptive augmentation." *Proceedings of the Web Conference 2021*.
>
> \[5\] Thakoor, Shantanu, et al. "Large-Scale Representation Learning on Graphs via Bootstrapping." *International Conference on Learning Representations* 2022.

---

> > ### Comment · Reviewer_xKF9 · 2025-08-03
> > **Thank you**
> >
> > Thank you for your response. However, several of my concerns remain unresolved.
> >
> > First, regarding the challenges mentioned in the manuscript, numerous augmentation-free methods for graph contrastive learning already exist. It is questionable whether the challenges you described are still present. What would be the result if these existing augmentation-free methods were directly applied to your task?
> >
> > Second, the current state of research in gene regulatory networks is unclear in the manuscript, which makes it difficult to judge the novelty of your work.
> >
> > [1] Augmentation-Free Self-Supervised Learning on Graphs, AAAI2022
> >
> > [2] SimGRACE: A Simple Framework for Graph Contrastive Learning without Data Augmentation, WWW2022

---

> > > ### Author Response · Authors · 2025-08-07
> > > **Comparison with Augmentation-Free Methods**
> > >
> > > Thank you for your valuable feedback and for allowing us to clarify our contributions. We address your remaining concerns below.
> > >
> > > ---
> > >
> > > ### **Answer 1: Comparison with Augmentation-Free Methods**
> > >
> > > Thank you for this critical question. We would first like to emphasize the **core philosophical difference** between our work and augmentation-free methods.
> > >
> > > The goal of SupGCL is not to avoid augmentations. Instead, our method is designed to leverage augmentations by directly learning from their topological changes. While augmentation-free GCL aims to prevent the negative effects of topological changes by designing contrastive frameworks that do not require them, SupGCL takes a different path. Our approach sublimates these topological changes into a **biologically meaningful supervisory signal** by using real-world gene knockdown data as the ground truth.
> > >
> > > In other words, while augmentation-free methods seek to avoid "incorrect" noise, SupGCL aims to **directly teach the model what constitutes a "correct" biological perturbation**. We believe this fundamental difference in philosophy is the primary reason for our method's superior performance in the experiments. This is substantiated by our comparisons with augmentation-free methods like GAE and SGRL, where our approach shows significant improvements.
> > >
> > > Regarding the specific methods you mentioned (AFGRL \[1\], SimGRACE \[2\]), we appreciate you bringing them to our attention. These methods are indeed important milestones in the evolution of augmentation-free GCL. This research area has progressed from introducing bootstrapping to graph contrastive learning (e.g., BGRL, which is not strictly augmentation-free), to combining bootstrapping with graph reconstruction (AFGRL), and simplifying perturbations with Gaussian noise (SimGRACE).
> > >
> > > As of early 2025, the state-of-the-art in this domain is widely considered to be SGRL (NeurIPS 2024), which enhances the bootstrap paradigm with representation scattering. The SGRL paper itself demonstrates its superiority over previous methods, including BGRL and AFGRL. For this reason, we selected SGRL as the representative state-of-the-art for augmentation-free GCL in our manuscript to ensure a fair and stringent comparison.
> > >
> > > However, to directly address your concern, we have conducted additional experiments comparing SupGCL with AFGRL and SimGRACE on key downstream tasks. The results, shown below, further confirm the superiority of our proposed method.
> > >
> > > | **Method** | **Hazard Prediction (C-index, Breast Cancer)** | **Gene Function Prediction (Subset Accuracy, Breast Cancer)** |
> > > | :--- | :---: | :---: |
> > > | AFGRL | 0.616 ± 0.037 | 0.240 ± 0.026 |
> > > | SimGRACE | 0.638 ± 0.032 | 0.228 ± 0.044 |
> > > | **SupGCL (Ours)** | **0.650 ± 0.059** | **0.243 ± 0.052** |
> > >
> > >
> > > **References**
> > >
> > > [1] Lee, G., & Lee, J. (2022). Augmentation-Free Self-Supervised Learning on Graphs. *AAAI*.
> > >
> > > [2] Xia, J., et al. (2022). SimGRACE: A Simple Framework for Graph Contrastive Learning without Data Augmentation. *WWW*.

---

> > > > ### Author Response · Authors · 2025-08-07
> > > > **Novelty and Positioning in Gene Regulatory Network Research**
> > > >
> > > > ### **Answer 2: Novelty and Positioning in Gene Regulatory Network Research**
> > > >
> > > >
> > > > Thank you for this insightful comment. It highlights the need to better contextualize our work. We agree that clarifying our position within the broader field of GRN research is crucial. **Should our manuscript be accepted, we would be pleased to incorporate this context into the Introduction and Related Work sections for the camera-ready version to further enhance the paper's value.**
> > > >
> > > > Current GRN research can be viewed as a continuous pipeline: from network estimation (inference) to **representation learning** on the inferred networks, and finally to applications in biomedical downstream tasks. Our work is sharply focused on the second stage: improving representation learning from pre-constructed GRNs.
> > > >
> > > > This stage is critically important, as high-quality representations directly impact the performance of vital clinical tasks like gene function prediction, cancer driver identification, and patient risk stratification. Recent advancements in this area have followed two main streams [4]: (1) integrating multi-modal data (e.g., MuSe-GNN [3]) and (2) adapting state-of-the-art graph representation learning techniques to GRNs . Our work belongs to the latter.
> > > >
> > > > While many recent studies have improved performance by upgrading the **encoder architecture** (e.g., using GAT in CancerGATE [5] or Graph Transformers in MuSe-GNN [3]), the fundamental approach to contrastive learning itself has remained generic.
> > > >
> > > > The core novelty of our work lies in innovating the **learning objective itself**, not just the encoder. We propose a paradigm shift for GRN representation learning with the following contributions:
> > > >
> > > > 1.  **Innovation in the Learning Objective**: For the first time, we have formulated an **experimentally verifiable biological perturbation** (gene knockdown) as a direct supervisory signal for GCL. This brings biological interpretability and validity to the augmentation process, which has largely been a black box.
> > > >
> > > > 2.  **Theoretical Generalization**: We provide a unified probabilistic framework that **theoretically generalizes existing GCL methods**, which emerge as special cases of our SupGCL formulation. This contributes to a more systematic understanding of the field.
> > > >
> > > > 3.  **Superiority**: We demonstrate that SupGCL consistently outperforms all baselines, including the SOTA augmentation-free method (SGRL), across a comprehensive benchmark of **13 downstream tasks and 3 cancer types**.
> > > >
> > > > In summary, where previous work addressed the challenge of "non-biological augmentations" by avoiding them or by improving encoder capacity, we tackle the problem head-on by proposing a new approach: **learning directly from the biological augmentations themselves**. This is why we are confident that our work represents a significant and novel contribution to the field of representation learning for gene regulatory networks.
> > > >
> > > > **References**
> > > >
> > > > [3] Liu, T., et al. (2023). MuSe-GNN: Learning unified gene representation from multimodal biological graph data. *NeurIPS*.
> > > >
> > > > [4] Zohari, Payam, and Mostafa Haghir Chehreghani. "Graph Neural Networks in Multi-Omics Cancer Research: A Structured Survey." arXiv preprint arXiv:2506.17234 (2025).
> > > >
> > > >
> > > > [5] Jung, S., et al. (2024). CancerGATE: Prediction of cancer-driver genes using graph attention autoencoders. *Computers in Biology and Medicine*.

---

### Official Review · Reviewer_LmUN · 2025-06-29

**Clarity:** 3
**Significance:** 2
**Originality:** 2
**Rating:** 4
**Confidence:** 4

**Summary:**

This paper introduces a novel graph contrastive learning (GCL) framework, SupGCL, specifically designed for the graph representation learning of gene regulatory networks (GRNs). SupGCL addresses a key challenge encountered when applying GCL methods to GRN representation learning: artificial perturbations can disrupt the heterogeneous characteristics of GRNs, thereby hindering the learning of effective graph representations. SupGCL constructs a Supervised Augmentation Model to impose constraints on the reference distribution in the loss function, and performs representation learning from both the node level and the graph level, thereby enhancing the overall learning performance. Experimental results demonstrate that SupGCL consistently outperforms conventional graph contrastive learning approaches across all downstream tasks.

**Questions:**

1. Regarding the avoidance of the trivial solution, I would like to ask whether you have considered adopting strategies similar to those used in SimSiam or MoCo to prevent model collapse, such as the stop-gradient technique or the momentum encoder.

2. Did you study how the amount of pretraining data affects model performance? Does using more data always lead to better results?

3. In addition to node-level and graph-level downstream tasks, have you considered using edge-level tasks to evaluate the model's performance? For example, link prediction tasks such as GRN inference.

4. For the comparison methods, why did you choose to change the model architecture to Graph Transformer instead of keeping the original network structure, such as GCN or GAT in GRACE.

**Ethical Concerns:**

["NO or VERY MINOR ethics concerns only"]

**Final Justification:**

My concerns were addressed, hence I raised score.

**Limitations:**

Yes.

**Paper Formatting Concerns:**

No concerns.

**Quality:**

2

**Strengths And Weaknesses:**

Strengths:
1. The authors consider a biologically motivated augmentation method, gene knockdown, and use the resulting augmented graphs to construct positive and negative sample pairs for graph contrastive learning.

2. The authors model the probability distribution from the perspective of the entire graph embedding space and integrate the reference model with the target model. This constitutes one of the key innovations of the work, proposing a novel paradigm for graph contrastive learning.

3. To prevent model collapse, where the model may converge to a trivial solution, the authors theoretically show that introducing a conditional probability distribution and assuming independence between nodes and augmentation operations in the reference distribution can effectively avoid such trivial solutions.

Weaknesses:
1. Although the authors propose the Supervised Augmentation Model, they do not provide ablation experiments to validate its effectiveness.

2. The motivation behind proposing the Supervised Augmentation Model is not clearly explained. Merely justifying it from the perspective of retaining biological characteristics is insufficient; the authors should also elaborate on why modifying the reference distribution in traditional GCL is necessary.

3. The temperature parameter in contrastive learning typically has a significant impact on model performance. However, the authors lack a thorough analysis of key hyperparameters; for instance, they do not present experiments with multiple settings to illustrate how the temperature parameter affects model performance.

4. When selecting baseline methods for comparison, the authors do not include state-of-the-art GNN-based approaches specifically designed for gene regulatory network (GRN) tasks. Moreover, the selected methods, such as GAE and GraphCL, are relatively outdated, and the number of comparison methods is also insufficient, leading to a lack of convincing experimental validation.

5. Regarding Result 2, the experimental results provided by the authors do not clearly demonstrate the superiority of SupGCL. More compelling evidence should be presented to show that the latent representations learned by SupGCL are of higher quality than those obtained by the comparison methods. It is difficult to assess the quality of the learned latent representations based solely on visual inspection. The authors are encouraged to include more quantitative metrics to enhance the objectivity and persuasiveness of the evaluation.

---

> ### Author Rebuttal · Authors · 2025-07-31
>
> **Response to Weakness 1: Ablation Study**
> - As you rightly point out, an ablation study of the proposed method is critically important. Here, we clarify that the ablation in our study corresponds to using the GRACE model, which is an unsupervised, node-level graph contrastive learning baseline. GRACE employs the node-level loss given in Equation (4) and selects augmentations uniformly at random—i.e., without our supervised augmentation selection. Consequently, GRACE can be viewed as our method with no biological “teacher,” sampling augmentations from a uniform distribution.
> - Moreover, the relationship between the ablation (GRACE) and our SupGCL method is formally established in Corollary 1. The result of Corollary 1 indicates that when increasing the augmentation temperature parameter $\tau_a$, our proposed method SupGCL continuously transitions to node-level unsupervised graph contrastive learning GRACE. This implies that as $\tau_a$ increases, the performance of our proposed method approaches that of the ablation study version GRACE.
> - Sine we had not clearly demonstrated how performance transitions continuously from the ablation model as the augmentation temperature parameter \(\tau_a\) varies, we present those results below. Although there is some divergence from the GRACE results, for the node‐level task (BP) we confirm that increasing \(\tau_a\) brings performance closer to that of GRACE, consistent with the mathematical proof in Corollary 1. This also shows that, starting from the ablation experiment of node‐level unsupervised graph contrastive learning, our proposed method achieves continuously improving performance by introducing a supervised signal through augmentation.
>
> | $\tau_a$ Setting            | Hazard (c-index)    | BP (Subset Accuracy) |
> |-------------------------------|--------------------:|---------------------:|
> | 0.1                           | 0.670 ± 0.078       | 0.262 ± 0.035       |
> | 0.25 (paper setting)          | 0.650 ± 0.059       | 0.243 ± 0.052       |
> | 0.5                           | 0.648 ± 0.053       | 0.261 ± 0.034       |
> | 1.0                           | 0.640 ± 0.056      | 0.244 ± 0.042       |
> | 2.0                           | 0.656 ± 0.060       | 0.237 ± 0.024       |
> | GRACE         | 0.642 ± 0.064       | 0.230 ± 0.051       |
>
> ---
>
> **Response to Weakness 2: Necessity of Introducing a Supervision Signal in Augmentation**
> - The motivation for incorporating a biological “teacher” via knock-down data is to further enhance downstream task performance beyond what standard self-supervised learning can achieve. Building on responses to Weakness 1, our method demonstrates superior performance in downstream tasks when compared with GRACE, which corresponds to biologically unsupervised learning in this research (Tables 2 & 3 in the main text).
>
> ---
>
> **Response to Weakness 3: Analysis of the Temperature Parameter’s Effect**
> - In our study, the temperature \(\tau_a\) was treated as a hyperparameter and was learned automatically via validation (see Appendix E.1).
> - However, as you suggest, we did not explicitly illustrate how varying \(\tau_a\) impacts performance. The ablation table above makes clear that, as \(\tau_a\) increases, performance shifts continuously from the GRACE baseline toward our SupGCL method.
>
> ---
>
> **Response to Weakness 4: Comparison with GRN-Specialized Representation Learning Models**
> - We agree that comparing against methods specifically designed for GRN representation learning is important. Most recent GRN-focused work has moved toward multimodal architectures, and pure GNN-only methods remain relatively scarce ([2], Table 2).
> - Below, we report downstream task performance when using Gene2Vec [1] as a baseline. Our SupGCL method outperforms Gene2Vec on both the graph-level (Hazard) and node-level (BP) tasks. While GAE and GraphCL are not the latest techniques, GAE remains a strong and widely cited baseline in this domain. SGRL, by contrast, is a state-of-the-art graph contrastive learning method introduced at NeurIPS 2024, and it demonstrates superior node classification performance. Thus, we believe the chosen comparators are appropriate.
>
> | Method               | Hazard (c-index, Breast) | BP (Subset Accuracy, Breast) |
> |----------------------|-------------------------:|-----------------------------:|
> | Gene2Vec             | 0.602 ± 0.042            | 0.236 ± 0.041               |
> | **SupGCL**    | 0.650 ± 0.059            | 0.243 ± 0.052               |
>
> ---
>
> **Response to Weakness 5: Quantitative Evaluation of the Embedding Space**
> - We agree that quantitative evaluation in the learned embedding space is crucial. To this end, we performed k-means clustering (with \(k=3\) for the BP task and \(k=5\) for the subtype classification task) on embeddings of breast cancer patients. We then measured similarity between cluster assignments and the ground-truth labels using Adjusted Rand Index (ARI) and Normalized Mutual Information (NMI). The results are shown below; SupGCL achieves the highest scores on both metrics.
>
> | Method     | BP of BREAST [ARI] | BP of BREAST [NMI] | Subtype of BREAST [ARI] | Subtype of BREAST [NMI] |
> |------------|-------------------:|-------------------:|------------------------:|------------------------:|
> | GAE        | 0.0010             | 0.0028             | 0.0204                  | 0.0282                 |
> | GRACE      | –0.0030            | 0.0066             | 0.1103                  | 0.1343                 |
> | GraphCL    | 0.0063             | 0.0028             | 0.0544                  | 0.0703                 |
> | SGRL       | 0.0105             | **0.0184**         | 0.0588                  | 0.0704                 |
> | **SupGCL** | **0.0272**         | 0.0036             | **0.1176**              | **0.1502**             |
>
> ---
>
> ### Response to Question 1: Measures against Model Collapse
> - In the current work, we did not employ any explicit anti-model collapse techniques (e.g., EMA, stop-gradient) because the training dynamics of our SupGCL method remained qualitatively stable.
> - To verify this, we added an Exponential Moving Average (EMA) of model parameters and evaluated downstream task performance. As shown below, incorporating EMA yields virtually identical results, indicating that explicit collapse-prevention mechanisms are unnecessary in our framework.
>
> | Setting          | Hazard (c-index)    | BP (Subset Accuracy) |
> |------------------|--------------------:|---------------------:|
> | SupGCL           | 0.650 ± 0.059      | 0.243 ± 0.052       |
> | SupGCL + EMA     | 0.644 ± 0.046      | 0.242 ± 0.036       |
>
> ---
>
> ### Response to Question 2: Scaling of the Pre-training Model
> - Investigating the scalability of the pre-training model is indeed important. We conducted additional experiments by reducing the number of pre-training samples to \(1/2\) and \(1/4\) of the original size, and measured downstream performance on the breast cancer dataset.
> - As shown below, the graph-level (Hazard) task exhibits a consistent upward trend with increasing data size. In contrast, the node-level (BP) task is largely insensitive to sample size, likely because node-level GCL methods can effectively learn node features even when the number of graphs is small (sometimes as few as one).
>
> | Sample Setting                    | Hazard (c-index)    | BP (Subset Accuracy) |
> |-----------------------------------|--------------------:|---------------------:|
> | Original (N = 1092)      | 0.650 ± 0.059      | 0.243 ± 0.052       |
> | 1/2 Sample              | 0.640 ± 0.040      | 0.243 ± 0.026       |
> | 1/4 Sample              | 0.631 ± 0.045      | 0.247 ± 0.038       |
>
> ---
>
> ### Response to Question 3: Application to GRN Inference (Edge-Level Tasks)
> - We have performed additional experiments on link prediction as an edge-level task. Our SupGCL method outperforms existing baselines, demonstrating its effectiveness beyond node- and graph-level tasks.
>
> | Method           | Accuracy           | F1 Score           |
> |------------------|-------------------:|--------------------:|
> | w/o pre-train    | 0.730 ± 0.021     | 0.843 ± 0.014      |
> | GAE              | 0.743 ± 0.024     | 0.846 ± 0.014      |
> | GraphCL          | 0.714 ± 0.031     | 0.824 ± 0.022      |
> | GRACE            | 0.757 ± 0.031     | 0.848 ± 0.016      |
> | SGRL             | 0.741 ± 0.032     | 0.835 ± 0.023      |
> | **SupGCL**       | **0.763 ± 0.028** | **0.855 ± 0.018**  |
>
> ---
>
> ### Response to Question 4: Validity of the GNN Architecture
> - As noted, the choice of GNN encoder is crucial. We selected the Graph Transformer for all experiments because the study in [2] (Appendix E.1) found that Graph Transformer outperforms other GNN encoders—GCN, GAT, TransformConv, SURGL, GPS, and GRACE—across different modalities (scRNA-seq, scATAC-seq, spatial transcriptomics). In particular, GCN- and GAT-based models struggled to capture gene functional similarity across datasets.
> - It is important to note that their study integrates multiple modalities, whereas our work focuses solely on GRN data. We further confirmed that replacing the encoder in GRACE with GAT leads to degraded downstream performance:
>
> | Model Configuration                                | Hazard (c-index)    | BP (Subset Accuracy) |
> |----------------------------------------------------|--------------------:|---------------------:|
> | GRACE (Graph Transformer, as in paper)            | 0.642 ± 0.064      | 0.230 ± 0.051       |
> | GRACE (with GAT encoder)                           | 0.632 ± 0.038      | 0.241 ± 0.040       |
>
> ---
>
> ### References
> 1. Du, J., Jia, P., Dai, Y. *et al.* (2019). Gene2vec: distributed representation of genes based on co-expression. *BMC Genomics*, 20, 82.
> 2. Liu, T., *et al.* (2023). Muse-gnn: Learning unified gene representation from multimodal biological graph data. *Advances in Neural Information Processing Systems*, 36, 24661–24677.

---

> > ### Comment · Reviewer_LmUN · 2025-08-04
> > **Official comments**
> >
> > Thank you for your response. The author addressed most of my concerns, but there are now many gene regulatory network inference methods based on supervised learning, including those based on graph neural networks and contrastive learning. The author did not compare the algorithm with these advanced methods, making it difficult to evaluate the performance of the algorithm.

---

> > > ### Author Response · Authors · 2025-08-07
> > > **Appreciation for the Insightful Comment: Clarifying Our Focus on GRN Representation Learning**
> > >
> > > Dear Reviewer,
> > >
> > > We would like to extend our sincere gratitude to the reviewer for this insightful and valuable comment. We certainly agree that a number of advanced supervised methods for GRN inference have recently been developed, and we appreciate the opportunity to clarify our work in light of this important point.
> > >
> > > ---
> > >
> > > ### Clarification of Research Scope
> > >
> > > It is our understanding that research concerning Gene Regulatory Networks (GRNs) can be generally categorized into two distinct stages:
> > >
> > >   * **Stage 1: GRN Inference:** This field of study is dedicated to inferring the regulatory relationships from gene expression data, with the objective of constructing the network structure itself.
> > >   * **Stage 2: GRN Representation Learning:** This area, in contrast, **presumes the existence of a pre-constructed GRN**. Its objective is to learn meaningful, low-dimensional representations (embeddings) from the network's topology and features, which can then be utilized for various downstream biological tasks.
> > >
> > > Our proposed methodology, **SupGCL, has been explicitly developed for Stage 2**. The central purpose of our investigation is to employ biologically plausible augmentations—specifically, those derived from authentic gene knockdown experiments—as a form of supervision to learn superior graph representations from a given GRN.
> > >
> > > In this context, the baseline methods selected for our study (GAE, GraphCL, GRACE, and SGRL) represent **standard and highly competitive approaches within the domain of graph representation learning (Stage 2)**. We believe they constitute the most suitable benchmarks for evaluating the primary contribution of our work.
> > >
> > > -----
> > >
> > > ### Comparison with GRN Inference Methods
> > >
> > > Conversely, the supervised GRN inference methods to which the reviewer kindly referred (such as GCLink[1]) are primarily situated within **Stage 1**. These methodologies generally diverge from our own in their primary objectives and underlying assumptions:
> > >
> > >   * **Objective**: Their principal aim is to **infer** previously unknown regulatory connections to build a network.
> > >   * **Supervision**: They often rely on **ground-truth information**, such as experimentally validated transcription factor-target interactions, for supervision.
> > >
> > > Our approach, however, is designed to be applicable to statistical causal networks from patient-derived data where such ground-truth information is typically unavailable. It is also intended to handle relationships among a broader set of genes. For these reasons, a direct comparison based on the singular metric of "inference performance" would be challenging, given the differences in research scope.
> > >
> > > -----
> > >
> > > ### Additional Experiment: Link Prediction Task
> > >
> > > Nonetheless, we fully agree that evaluating our learned representations from multiple perspectives and comparing them with state-of-the-art supervised methods is of the utmost importance. To further demonstrate the robustness and quality of the embeddings produced by SupGCL, we conducted an additional experiment to compare how effectively our Stage 2 method can be applied to a link prediction task, which is characteristic of Stage 1. In this experiment, we deliberately masked a subset of edges from the statistical GRNs and assessed the model's ability to predict these missing links. For comparison, we used GCLink[1], a representative state-of-the-art method in GRN inference.
> > >
> > > In this experiment, we deliberately masked a subset of edges from the statistical GRNs and assessed the capability of SupGCL to predict these missing links, comparing its performance against GCLink, a representative GRN inference method.
> > > The evaluation was conducted using the same experimental protocol as described in our response to your third question (Q3) of the previous comment.
> > > The results of this analysis are presented below:
> > >
> > > | Method        | Accuracy        | F1 score        |
> > > |:--------------|:----------------|:----------------|
> > > | GCLink[1]        | 0.756 ± 0.031   | 0.853 ± 0.018   |
> > > | **SupGCL (ours)** | **0.763 ± 0.028** | **0.855 ± 0.018** |
> > >
> > > It is important to note that the target for this GRN inference task was the statistical causal network used throughout our study, rather than the ground-truth transcriptional network typically used in experiments for GCLink.
> > >
> > > These results suggest that SupGCL demonstrates a performance level that is **highly competitive with, and even marginally exceeds, a specialized inference method in a link prediction task**, despite this not being its primary design objective.
> > >
> > > We believe these findings offer compelling evidence that our proposed framework—utilizing biological perturbations for supervised contrastive learning—is exceptionally effective at generating rich, high-quality embeddings that adeptly capture the fundamental structural properties of the network.
> > >
> > > [1] Yu, Weiming, et al. "GCLink: a graph contrastive link prediction framework for gene regulatory network inference." Bioinformatics 41.3 (2025): btaf074.

---

### Official Review · Reviewer_iYjm · 2025-07-02

**Clarity:** 2
**Significance:** 2
**Originality:** 3
**Rating:** 4
**Confidence:** 2

**Summary:**

This paper proposes SupGCL for learning biologically meaningful representations of GRNs. It improves standard GCL by supervising artificial graph augmentations (simulated gene knockdowns) with real biological data from experimental knockdowns. Using a shared graph neural network, it aligns node-level and graph-level embeddings between fake and real knockdowns via KL-divergence-based contrastive losses, enabling more faithful modeling of gene interactions.

**Questions:**

Expanding on the weaknesses described above, I propose the following questions which the authors may consider answering.

1.  It would be good to do some analysis where you cluster the embeddings and see if those clusters line up with known biological groups or gene ontology terms.
2. Can we add a way to weight the knockdown graphs based on how trustworthy they are, or filter out the noisy ones?
3. Is it possible to check how well the model works on new genes or cell types and if there are any risks of overfitting to knockdown data from few cell lines?

**Ethical Concerns:**

["NO or VERY MINOR ethics concerns only"]

**Final Justification:**

My concerns were addressed, hence I raised score but my confidence of assessment is lower compared to other reviewers.

**Limitations:**

Yes

**Quality:**

3

**Strengths And Weaknesses:**

***Strengths***:
1. SupGCL uses actual gene knockdown data to guide how the graph is augmented, instead of just applying random or hand-crafted changes. This makes sense from biological perspective.
2. The method combines losses that look at both individual nodes and the whole augmented graph using perturbation data. This, to my knowledge, is novel use of perturbation data to design augmentations for GCL.

***Weaknesses***:
1. The paper claims that the propose augmentation strategy helps improve biological realism. Is it possible to verify this more directly other than using the results  on downstream tasks to support this hypothesis.
2. SupGCL assumes all the knockdown graphs are equally reliable, but in biological contexts some knockdown experiments may have more relevance than others. Can the framework handle such scenarios?
3. Since the model learns to match artificial and real knockdowns, is it possible that it can learn/memorize batch effects from knockdown data in case they all come from a few cell lines?

---

> ### Author Rebuttal · Authors · 2025-07-31
>
> Dear Reviewer,
>
> Thank you very much for your valuable and constructive feedback on our manuscript. We have carefully considered your comments and have revised our paper accordingly.
> # Q1.
> We conducted additional analyses on (i) quantitative evaluation and (ii) biological interpretation related to the embedding space.
>
> (i) Quantitative evaluation of the embedding
> We compared clustering derived from the embeddings to ground-truth labels in two tasks:
> the BP task (upper panel of Fig. 3) and subtype classification (lower panel of Fig. 3).
> For each method, we ran k-means with the same number of clusters as the task (3 for BP, 5 for subtype) and measured alignment using ARI and NMI.
>
> SupGCL achieves the highest ARI in both tasks, indicating stronger cluster-level separability.
>
> |        | BP of BREAST [ARI] | BP of BREAST [NMI] | Subtype of BREAST [ARI] | Subtype of BREAST [NMI] |
> |:------:|:------------------:|:------------------:|:-----------------------:|:-----------------------:|
> | GAE    | 0.0010             | 0.0028             | 0.0204                  | 0.0282                  |
> | GRACE  | -0.0030            | 0.0066             | 0.1103                  | 0.1343                  |
> | GraphCL| 0.0063             | 0.0028             | 0.0544                  | 0.0703                  |
> | SGRL   | 0.0105             | 0.0184         | 0.0588                  | 0.0704                  |
> | SupGCL | 0.0272     | 0.0036             | 0.1176              | 0.1502              |
>
> (ii) Biological interpretation via GO and KEGG enrichment
>
> For each method's gene embeddings, we performed k-means clustering (K = 7) and then conducted cluster-wise enrichment analysis with g:Profiler for GO:BP and KEGG. The number of clusters (K = 7) was selected using the elbow method applied to the within-cluster sum of squares, evaluated across methods.
> Clusters with fewer than five genes were excluded from enrichment, and terms/pathways are reported in order of statistical significance.
>
> a. Clustering results
> | Cluster  | GAE        | GraphCL   | GRACE     | SGRL      | SupGCL   |
> |----------|------------|-----------|-----------|-----------|----------|
> | Cl 0     | 53    | 482  | 73   | 658  | 251 |
> ...
> | Gini Index | 0.375 | 0.632 | 0.594 | 0.699 | 0.334 |
>
> SupGCL and GAE yield comparatively lower Gini indices, indicating more balanced cluster sizes than other baselines.
>
> b. Gene Ontology enrichment
> b.1 Results (GO:BP; Top 5 per cluster)
>
> Next, we present the significantly enriched GO terms (Top 5) for each method and each cluster from the GO enrichment analysis. As comparison methods to ours, we include the reconstruction-based GAE, which is frequently used in GRN representation learning, and  GRACE, which serves as an ablation of our approach. Clusters with fewer than five genes were excluded. Terms are ranked by significance. Note that “Cl x” refers to cluster IDs within each method; no alignment between clusters across different methods is implied.
>
> | Cluster | GAE | GRACE | SupGCL |
> |---------|-----|-------|--------|
> | Cl 0 | 1. Negative regulation of reactive oxygen species metabolic process2. Negative regulation of intrinsic apoptotic signaling pathway3. Intrinsic apoptotic signaling pathway4. Regulation of reactive oxygen species metabolic process5. Lymphoid progenitor cell differentiation | 1. Intrinsic apoptotic signaling pathway2. Regulation of programmed cell death3. Intrinsic apoptotic signaling pathway in response to DNA damage4. Apoptotic process5. Programmed cell death | 1. Organelle organization2. Macromolecule localization3. Chromosome organization4. Catabolic process5. Cellular response to stress |
> ...
> "—" = no significant pathways
>
> SupGCL and GAE recovered significant (FDR < 0.05) functional modules in 6/7 clusters, whereas GRACE did so in 4/7.
>
> b.2
> We discussed the cluster characteristics for each method, for example:
> - SupGCL.Cl 0 aggregates organelle organization, macromolecule localization, chromosome organization, catabolic process, and cellular response to stress, suggesting a mixed structural/transport and stress/turnover module.
> The results of the analysis of all clusters and all comparison methods are presented in a paper.
> The results show that SupGCL and GAE are superior in terms of interpretability via biological module separation; notably, SupGCL captures the hierarchy of autophagy (regulation vs execution).
>
> c. KEGG pathway enrichment
>
> c.1 Results (KEGG; Top 5 per cluster)
> | Cluster | GAE | GraphCL | GRACE | SGRL | SupGCL |
> |---------|-----|---------|-------|------|--------|
> | Cl 0 | 1. Amino sugar and nucleotide sugar metabolism | — | 1. Kaposi sarcoma–associated herpesvirus infection2. Epstein–Barr virus infection3. Hepatitis B4. Colorectal cancer5. Human immunodeficiency virus 1 infection | — | 1. Mismatch repair |
> ...
>
> c.2 Method-specific characteristics
> Similar to GO and KEGG enrichment, we described the cluster characteristics of each method.
>
> d. Summary
> SupGCL, based on breast cancer samples, clearly separates key pathways: regulatory vs. executional autophagy, hormone signaling/ER stress (Cl 6), stress–metabolic regulation (Cl 5), and DNA repair (Cl 0).
> It captures two distinct therapeutic axes:
> Hormone signaling → ER stress → autophagy execution
> p53/FoxO → metabolic shift → autophagy regulation
> Compared to GAE and GRACE, SupGCL offers more precise, interpretable modules specific to breast cancer.
>
>
> # Q2.
>
> This study used GRN edge contribution scores as GNN edge features, can be regarded as reliability. To test robustness against noise in estimated GRNs, we conducted additional experiments using Bayesian network inference, which is prone to errors from initial parameters. To reduce this noise, we estimated 1,000 GRNs and filtered edges by appearance frequency, using a 5% threshold in the main results and providing comparisons with 3% and 10% thresholds in the table below.
>
> SupGCL:
> | Setting             | Hazard | BP  |
> |---------------------|:------------------------:|:-----------------------------:|
> | 3% cutoff           | 0.665 ± 0.059            | 0.238 ± 0.029                 |
> | 5% cutoff(original) | 0.650 ± 0.059            | 0.243 ± 0.052                 |
> | 10% cutoff          | 0.646 ± 0.039            | 0.244 ± 0.029                 |
>
> Small variations across cutoffs indicate stable pretraining despite GRN uncertainty, supporting robustness to GRN noise.
>
> # Q3A. novel genes
>
> Our current setup does not explicitly handle genes entirely unseen during pretraining. Contrastive learning is conducted over the observed GRN; extending to truly novel genes (e.g., zero-shot via neighborhood inference) is beyond the present scope.
>
> # Q3B. novel cell lines
>
> We considered three settings:
> 1. Cross-domain during pretraining (teacher GRN $H_a$ from one cancer type applied to patient GRN $\mathcal{G}$ from another).
>    SupGCL’s contrastive signal depends on differences between perturbations (embeddings of knockdowns $a, b$) modeled by the teacher $p_\phi(b \mid a)$, which may transfer across cancer types. We tested Breast-derived $\mathcal{H}_a$ against Lung/Colon $\mathcal{G}$, comparing to matched-domain references.
>
>    | Configuration | Task (Hazard / BP) | Hazard | BP  |
>    |:------------- |:------------------- |:----------------:|:--------------------:|
>    |  Lung -> Breast (Cross-domain) | Graph / Gene-function | 0.633 ± 0.069 | 0.270 ± 0.060 |
>    |  Lung -> Lung (Reference)      | Graph / Gene-function | 0.627 ± 0.051 | 0.282 ± 0.037 |
>    |  Colon -> Breast (Cross-domain)| Graph / Gene-function | 0.687 ± 0.092 | 0.257 ± 0.044 |
>    |  Colon -> Colon (Reference)    | Graph / Gene-function | 0.698 ± 0.085 | 0.262 ± 0.030 |
>
>    Pretraining with Breast knockdowns achieves comparable performance to matched-domain references, suggesting some cross-domain transferability at pretraining time.
>
>
> 2. Cross-domain during fine-tuning.
>    Fine-tuning requires paired GRN and labels for the target type. We fine-tuned on Breast using pretrained models from different cancer types.
>
>    | Pretraining / Fine-tuning                | Hazard  | BP  |
>    |:-----------------------------------------|:-------------------------:|:-----------------------------:|
>    | Breast/Breast (original) | 0.650 ± 0.059      | 0.243 ± 0.052                 |
>    | Lung/Breast      | 0.654 ± 0.075            | 0.248 ± 0.043                 |
>    | Colon/Breast     | 0.632 ± 0.072            | 0.249 ± 0.030                 |
>
>    Lung → Breast transfers well; Colon → Breast shows degradation, indicating source–target mismatch matters for fine-tuning.
>
> 3. OOD prediction with a trained model.
>    We evaluated models trained (pretrained + fine-tuned) on Breast and then tested on Lung/Colon.
>
>    | Cancer type (Fine-tune / Predict) | Hazard  | BP  |
>    |:----------------------------------|:----------------:|:--------------------:|
>    | Breast/Lung                     | 0.6032           | 0.1634               |
>    | Lung/Lung (original)            | 0.627 ± 0.051    | 0.282 ± 0.037        |
>    | Breast/ Colon                    | 0.5826           | 0.1624               |
>    | Colon/Colon (original)          | 0.698 ± 0.085    | 0.262 ± 0.030        |
>
> Cross-cancer deployment shows clear performance drops, indicating limited OOD robustness in the current formulation.
>
> # Summary of Q3.
> SupGCL’s pretraining tolerates some domain mismatch, but fine-tuning and deployment across cancer types suffer. We acknowledge this as a limitation.
>
> ## Additional Analysis: Robustness to limited data
> To assess overfitting risk with few knockdown-derived teacher GRNs, we subsampled the Breast knockdown set (8793 → 4397 → 2199) and evaluated downstream performance.
> Although detailed results were presented in the paper,
> we observed no significant degradation even with one-quarter of the teacher data. Given a 975-gene GRN, the 1/4 setting corresponds to roughly ~2 experiments per gene, suggesting SupGCL does not readily overfit under realistic data availability.

---

> > ### Comment · Reviewer_iYjm · 2025-08-03
> >
> > Thank you for the responses! My questions have been addressed!

---

### Official Review · Reviewer_dd11 · 2025-07-04

**Clarity:** 3
**Significance:** 3
**Originality:** 3
**Rating:** 4
**Confidence:** 4

**Summary:**

This paper proposes SupGCL, a new supervised graph contrastive learning method tailored for gene regulatory networks (GRNs). Traditional graph contrastive learning approaches typically rely on artificial perturbations such as random node or edge removal for data augmentation, which are often biologically implausible when applied to GRNs. In contrast, SupGCL leverages real biological perturbations by incorporating gene knockdown experiment data as supervision to guide the learning of graph representations. The authors formulate this approach within a probabilistic KL divergence-based framework, demonstrating that existing unsupervised GCL methods are special cases of their formulation when supervision is removed. They apply SupGCL to GRNs constructed from cancer patient data and gene knockdown experiments across breast, lung, and colorectal cancer datasets. The model is evaluated on various downstream tasks, including gene function classification (node-level tasks), patient hazard prediction, and disease subtype classification (graph-level tasks). Experimental results show that SupGCL consistently outperforms state-of-the-art baselines in both node- and graph-level evaluations, suggesting its effectiveness in learning biologically meaningful GRN representations that can enhance predictive performance in biomedical applications.

**Questions:**

The proposed SupGCL framework uses gene knockdown data from the same cancer type as supervision. How does the method perform if knockdown data from other cancer types are used, or if certain knockdowns are unavailable for a target cancer type?

While the tables report means and standard deviations, were statistical significance tests conducted to compare SupGCL against baselines on downstream tasks?

The paper mentions that hyperparameters were tuned with Optuna, but it is unclear whether all baseline models underwent equivalent tuning efforts.

Beyond t-SNE visualizations, are there biological interpretations or validation of the learned embeddings, such as enrichment analyses or biological pathway coherence for gene-level representations?

Given the paper’s motivation around clinical and therapeutic applications, what are the practical considerations or limitations for deploying SupGCL in real clinical genomics pipelines (e.g. computational cost, data availability, required supervision)?

**Ethical Concerns:**

["NO or VERY MINOR ethics concerns only"]

**Final Justification:**

I have no more questions and will keep my rating.

**Limitations:**

The discussion of potential negative societal impacts could be expanded. For example, while the method advances biological understanding, its use in clinical decision-making without rigorous validation could risk misinterpretation of patient-specific predictions. Additionally, there could be implications regarding equitable access to such computational tools in genomics.

Including a brief reflection on these societal aspects would strengthen the ethical transparency of the paper without detracting from its scientific focus.





Ask ChatGPT

**Quality:**

3

**Strengths And Weaknesses:**

Strengths:
The paper demonstrates strong technical quality. It provides a clear mathematical formulation of SupGCL as a probabilistic extension of conventional GCL, incorporating supervised biological perturbations (gene knockdowns). The theoretical connection showing that existing GCL methods are special cases of SupGCL is rigorous and well-motivated. Empirical results are comprehensive, covering both node-level and graph-level tasks across multiple cancer types with meaningful biological datasets (TCGA and LINCS). The experimental setup, including pre-training and fine-tuning protocols, is sound, and performance improvements are consistently demonstrated over strong baselines such as GraphCL, GRACE, and SGRL.

The significance of this work is high. It addresses an important problem in biological network representation learning: the lack of biologically meaningful data augmentations in GCL, which limits applicability to real biological tasks. By incorporating gene knockdown data, SupGCL enables learning representations that are more aligned with biological realities, potentially benefiting downstream biomedical tasks such as disease subtyping, hazard prediction, and functional gene annotation. This approach could inspire broader integration of supervised perturbations in contrastive learning for other structured biological or clinical data domains.

Weaknesses:
Although the paper is technically solid, some details could be improved.

While the authors demonstrate performance gains, statistical significance testing across methods is not thoroughly discussed, which would strengthen claims of superiority.

Hyperparameter choices for SupGCL versus baselines are briefly mentioned but could be detailed further to rule out tuning bias.

While Theorem 1 is proved, the practical intuition behind some derivations (e.g. KL decomposition under independence assumptions) could be elaborated for accessibility.

A limitation in significance is that the evaluation is restricted to knockdown perturbations from the same cancer type as the patient GRNs. While the authors acknowledge this, it constrains the generalizability claims of SupGCL.

Practical integration into clinical pipelines remains speculative and is not discussed, which would strengthen its real-world relevance.

The paper assumes substantial familiarity with GCL literature and probabilistic formulations, which may limit accessibility to readers from computational biology backgrounds who are potential beneficiaries of this work. Certain sections, especially the derivation-heavy Method section, could benefit from intuitive summaries before formal equations to maintain readability.

While the incorporation of supervised biological perturbations is novel, the contrastive learning formulation itself is an extension rather than a fundamentally new learning paradigm. The broader originality is therefore dependent on the biological application rather than the machine learning method itself.

---

> ### Author Rebuttal · Authors · 2025-07-31
>
> Dear Reviewer,
>
> Thank you very much for your valuable and constructive feedback on our manuscript.
> Unfortunately, due to the character limit, we were unable to list all results. The complete results will be provided during additional rebuttal comments or paper revisions.
>
>
> # Response to Weakness
> In our reference model we assume independence between the node and the augmentation operator. This assumption naturally yields the node-level error term (first term in Eq. (9)), helps avoid trivial solutions, and clarifies the connection with prior methods (formalized in Lemma 1). We will add a concise intuitive explanation of this assumption and its role in the KL decomposition to the main text.
>
> # Q1
>
> Our long-term objective is cancer-type–agnostic GRN representation learning. SupGCL is scale-free with respect to specific gene identities and target knockdowns and thus can, in principle, be applied to diverse GRNs. In particular, the teacher model $p_\phi(b \mid a)$ requires only the difference between selected augmentations—i.e., the embedding difference between knockdown perturbations $a$ and $b$—which may be cancer-agnostic. To assess this, we conducted additional cross-domain experiments.
>
> (i) Cross-domain at pretraining time.
>
> We pretrained with knockdown-derived teacher GRNs from Breast and evaluated on patient GRNs  from Lung/Colorectal, comparing against matched-domain references.
>
>
> | Configuration | Task (Hazard / BP) | Hazard (c-index) | BP (Subset Accuracy) |
> |:------------- |:------------------- |:----------------:|:--------------------:|
> | Lung -> Breast (Cross-domain) | Graph / Gene-function | 0.633 ± 0.069 | 0.270 ± 0.060 |
> | Lung -> Lung (Reference)      | Graph / Gene-function | 0.627 ± 0.051 | 0.282 ± 0.037 |
> | Colorectal -> Breast (Cross-domain)| Graph / Gene-function | 0.687 ± 0.092 | 0.257 ± 0.044 |
> | Colorectal -> Colorectal (Reference)    | Graph / Gene-function | 0.698 ± 0.085 | 0.262 ± 0.030 |
>
> Using Breast knockdowns yields performance comparable to matched-domain references, suggesting feasibility of cross-domain pretraining given current data.
>
> (ii) Cross-domain at fine-tuning phase.
>
> Because fine-tuning only requires paired GRNs and labels for the target type, we can fine-tune even when the pretraining cancer type differs. We fine-tuned on Breast using models pretrained on each of three cancer types:
>
> | Pretraining / Fine-tuning | Hazard (c-index, Breast) | BP (Subset Accuracy, Breast) |
> |:--------------------------|:------------------------:|:----------------------------:|
> | Breast; Breast (original) | 0.650 ± 0.059 | 0.243 ± 0.052 |
> | Lung;  Breast           | 0.654 ± 0.075 | 0.248 ± 0.043 |
> | Colorectal; Breast        | 0.632 ± 0.072 | 0.249 ± 0.030 |
>
> Pretraining on Lung transfers well to Breast, whereas Colorectal shows degradation; thus, source–target mismatch is not negligible.
>
>
> (iii). Evaluation of Out-of-Distribution Performance of the Trained Model
>
> We evaluated the performance of the fine-tuned trained model on novel cancer types.
> The model was pretrained and fine-tuned using gene expression networks, knockdown experiment data, and label data from breast cancer patients. We then made predictions using data from other cancer types; the results are shown below.
>
> | Cancer type of fine-tuning / prediction | Hazard: graph-level task (c-index) | BP: node-level task (subset accuracy) |
> |:--------------------------------------|:----------------------------------:|:------------------------------------:|
> | Breast, Predict: Lung | 0.6032 | 0.1634 |
> | Lung, Predict: Lung (original) | 0.627 ± 0.051 | 0.282 ± 0.037 |
> | Breast, Predict: Colorectal | 0.5826 | 0.1624 |
> | Colorectal, Predict: Colorectal (original) | 0.698 ± 0.085 | 0.262 ± 0.030 |
>
> Because performance clearly degrades when using data from other cancer types, the current single-cancer-type pretrained model lacks robustness to novel cancer types.
> The above results indicate that, although there is no noticeable discrepancy in the pretraining phase across new cell types, performance degradation is observed during fine-tuning and evaluation of the trained model. These constitute limitations of the present study.
>
>
> # Q2
>
> We agree that reporting significance tests is informative. We conducted two-sample Student's t-tests comparing SupGCL with each of five baselines on downstream tasks, applying Bonferroni correction and a 5% significance level. Most comparisons did not reach significance; however, several node-level tasks did show significant differences. Below we list tasks/cancers/methods with significant differences (corrected p-values):
> We will include full per-task statistics (effect sizes, confidence intervals, and multiple-comparison procedures) in the appendix.
>
> # Q3
>
> All baselines, not only SupGCL, were tuned with Optuna. We harmonized search spaces across methods (details in Appendix E.1). Where baselines required additional method-specific hyperparameters, these were also optimized via Optuna under the same budget.
>
> # Q4
>
> As correctly noted by the reviewers, our original submission lacked sufficient biological analysis of the latent space. We therefore conducted additional enrichment analyses on the pretrained embeddings learned from the Breast Cancer GRN, using Gene Ontology (GO) and KEGG.
> We analyzed pretrained gene embeddings derived from the Breast Cancer GRN using k-means clustering (K = 7, selected by the elbow method on within-cluster SSE), followed by g:Profiler enrichment for GO:BP and KEGG; clusters with fewer than five genes were excluded and significance was assessed at FDR < 0.05.
>
> Clustering results
> | Cluster  | GAE       | GraphCL  | GRACE    | SGRL     | SupGCL  |
> |----------|-----------|----------|----------|----------|---------|
> | Cl 0     | 53        | 482      | 73       | 658      | 251     |
> ...
> | Gini Index | 0.375  | 0.632    | 0.594    | 0.699    | 0.334   |
>
> Observation. SupGCL and GAE yield comparatively lower Gini indices, indicating more balanced cluster sizes than other baselines.
>
>
> # a. Gene Ontology enrichment
> ## a.1 Results (GO:BP; Top 5 per cluster)
>
> | Cluster | GAE | GRACE | SupGCL |
> |---------|-----|-------|--------|
> | Cl 0 | 1. Negative regulation of reactive oxygen species metabolic process2. Negative regulation of intrinsic apoptotic signaling pathway3. Intrinsic apoptotic signaling pathway4. Regulation of reactive oxygen species metabolic process5. Lymphoid progenitor cell differentiation | 1. Intrinsic apoptotic signaling pathway2. Regulation of programmed cell death3. Intrinsic apoptotic signaling pathway in response to DNA damage4. Apoptotic process5. Programmed cell death | 1. Organelle organization2. Macromolecule localization3. Chromosome organization4. Catabolic process5. Cellular response to stress |
> ...
>
> Coverage. SupGCL and GAE recovered significant (FDR < 0.05) functional modules in 6/7 clusters, whereas GRACE did so in 4/7.
>
> ## b.2 Method-specific characteristics
>
> - SupGCL. Autophagy-related processes split into two modules: a regulatory cluster (Cl 3; regulation/positive regulation of (macro)autophagy) and an execution cluster (Cl 6; autophagy/macroautophagy/process utilizing autophagic mechanism). Cl 0 aggregates organelle organization, macromolecule localization, chromosome organization, catabolic process, and cellular response to stress, suggesting a mixed structural/transport and stress/turnover module. Cl 4 is dominated by positive regulation of biological process and developmental terms (e.g., system/nervous system development), indicating a differentiation/development module.
>
> # c. KEGG pathway enrichment
>
> ## c.1 Results (KEGG; Top 5 per cluster)
>
> | Cluster | GAE | GRACE | SupGCL |
> |---------|-----|-------|--------|
> | Cl 0 | 1. Amino sugar and nucleotide sugar metabolism | 1. Kaposi sarcoma–associated herpesvirus infection2. Epstein–Barr virus infection3. Hepatitis B4. Colorectal cancer5. Human immunodeficiency virus 1 infection | 1. Mismatch repair |
> ...
> Legend: "—" = no significant pathways
>
> ## c.2 Method-specific characteristics
>
> - SupGCL. Cl 0: Mismatch repair; Cl 5: FoxO/p53/Insulin resistance (stress–metabolic regulation); Cl 6: Protein processing in ER, Endocrine resistance, Estrogen signaling. The Cl 6 constellation is consistent with breast cancer context (endocrine resistance and ER stress).
>
>
> # d. Summary
> By repeating the above analysis on other clusters, we came to the following conclusions:
> SupGCL, applied to breast cancer–derived data, effectively identifies distinct biological modules. It separates autophagy into regulatory and executional clusters, and distinguishes endocrine resistance/ER stress from p53/FoxO-mediated stress and metabolic regulation. These findings reflect two key, non-overlapping pathways in therapeutic resistance. Compared to GAE and GRACE, SupGCL offers superior biological interpretability through clearer, cancer-specific module separation.
>
>
> # Q5
>
> The proposed SupGCL method requires knockdown data from target genes as supervision, which represented one of its major limitations. However, the results of the additional experiments described above suggest that pre-training can be performed using knockdown data from other cancer types, potentially eliminating this limitation.
> On the other hand, fine-tuning and evaluation of pre-trained models show performance degradation. This indicates that improving fine-tuning and prediction performance for arbitrary cancer types through pre-training of a single cancer type remains a significant limitation of this approach.
>
> Due to the architectural characteristics of the Graph Transformer used in this research, the computational complexity of SupGCL increases quadratically with the number of nodes (genes). When training with a larger number of genes, the associated computational requirements may become a substantial challenge.

---

> ### Comment · Area_Chair_h7ib · 2025-08-05
>
> Dear Reviewer dd11,
>
> Please help go through the rebuttal and participate in discussions with authors. If possible, you can also take a look at the review comments from other reviewers. Thank you!
>
> Best regards,
> AC

---

### Note · Authors · 2025-08-13

We believe that we have thoroughly resolved the concerns raised by the reviewers through our additional experiments.
We will revise the manuscript based on the following responses.

# Biological Plausibility of Using Knockdown-Derived GRNs as Supervision (LmUN, xKF9)
Traditional methods for Gene Regulatory Network (GRN) analysis have predominantly focused on simple link prediction without representation learning or on augmentation-free techniques like GAE.
In contrast, our model, SupGCL, leverages real-world data to learn from the topological changes induced by augmentation, thereby enhancing learning performance.
Our approach uniquely elevates these topological changes into biologically meaningful supervisory signals by using actual gene knockdown data as supervision.

Our extensive experiments demonstrate that SupGCL significantly outperforms both augmentation-free methods (GAE, SGRL, AFGRL, and SimGRACE) and GRN-specific approaches (Gene2vec and GCLink), confirming the superiority of our methodology.

# Validity of the Feature Space (dd11, iYjm)
We have quantitatively confirmed that the GRN features embedded by our pre-trained model exhibit superior clustering performance at both the node and graph levels. Furthermore, additional enrichment analysis revealed that SupGCL effectively separates autophagy-related clusters and the ER stress pathway.
This result substantiates the excellent biological interpretability of the feature space learned by SupGCL.

# Application to Different Cancer Types (dd11)
In addressing concerns about SupGCL's cross-domain performance, our evaluation revealed that its performance is robust during the pre-training phase, even when supervised and patient-derived GRNs come from mismatched cancer types.
However, performance degrades if cancer types do not match during the fine-tuning and evaluation phases. This clarifies that while SupGCL is robustly cross-domain in pre-training, its performance is compromised by mismatched cancer types in later stages.

# Robustness to GRN Inference Accuracy (iYjm, LmUN)
Additional experiments confirmed our method's robustness.
Performance remained stable when we added noise to the GRNs and limited the number of supervised GRNs used for training.

# Scalability of the Proposed Method (LmUN)
For graph-level tasks, performance scaled with pre-training data, confirming our method's scalability.
Conversely, node-level task performance was constant, as it depends on the number of nodes per graph.

---

### Decision · Program_Chairs · 2025-09-17

**Decision:**

Reject

**Comment:**

This paper proposes SupGCL, a supervised graph contrastive learning framework that leverages gene knockdown data to guide biologically meaningful augmentations for gene regulatory networks (GRNs). However, the reviewers raise significant concerns that remain insufficiently addressed. In particular, the authors did not provide an in-depth discussion of prior research on GRN in related work section. Also, the comparisons are limited to generic GCL baselines, without including state-of-the-art approaches specifically designed for GRN inference, making it difficult to fairly assess the claimed improvements. The effectiveness of the supervised augmentation model is also not convincingly validated. As these issues are not well addressed in the rebuttal, I would like to recommend rejecting this paper.